# Involvement of a Cluster of Basic Amino Acids in Phosphorylation-Dependent Functional Repression of the Ceramide Transport Protein CERT

**DOI:** 10.3390/ijms23158576

**Published:** 2022-08-02

**Authors:** Asako Goto, Daichi Egawa, Nario Tomishige, Toshiyuki Yamaji, Kentaro Shimasaki, Keigo Kumagai, Kentaro Hanada

**Affiliations:** 1Department of Biochemistry and Cell Biology, National Institute of Infectious Diseases, Shinjuku-ku, Tokyo 162-8640, Japan; gotoa@niid.go.jp (A.G.); daichi.egawa@sums.ac.jp (D.E.); tyamaji@niid.go.jp (T.Y.); shimak@niid.go.jp (K.S.); kuma@niid.go.jp (K.K.); 2Department of Quality Assurance, Radiation Safety, and Information System, National Institute of Infectious Diseases, Shinjuku-ku, Tokyo 162-8640, Japan

**Keywords:** sphingolipid, phosphoinositide, lipid trafficking, endoplasmic reticulum, golgi, lipid transfer protein, sphingomyelin, *CERT1*, intragenic suppression

## Abstract

Ceramide transport protein (CERT) mediates ceramide transfer from the endoplasmic reticulum to the Golgi for sphingomyelin (SM) biosynthesis. CERT is inactivated by multiple phosphorylation at the serine-repeat motif (SRM), and mutations that impair the SRM phosphorylation are associated with a group of inherited intellectual disorders in humans. It has been suggested that the *N*-terminal phosphatidylinositol 4-monophosphate [PtdIns(4)P] binding domain and the *C*-terminal ceramide-transfer domain of CERT physically interfere with each other in the SRM phosphorylated state, thereby repressing the function of CERT; however, it remains unclear which regions in CERT are involved in the SRM phosphorylation-dependent repression of CERT. Here, we identified a previously uncharacterized cluster of lysine/arginine residues that were predicted to be located on the outer surface of a probable coiled-coil fold in CERT. Substitutions of the basic amino acids in the cluster with alanine released the SRM-dependent repression of CERT activities, i.e., the synthesis of SM, PtdIns(4)P-binding, vesicle-associated membrane protein-associated protein (VAP) binding, ceramide-transfer activity, and localization to the Golgi, although the effect on SM synthesis activity was only partially compromised by the alanine substitutions, which moderately destabilized the trimeric status of CERT. These results suggest that the basic amino acid cluster in the coiled-coil region is involved in the regulation of CERT function.

## 1. Introduction

Lipid transfer proteins (LTPs) play pivotal roles in the inter-organelle trafficking of lipids in eukaryotes [1,2,3,4]. The ceramide transport protein, CERT, a typical LTP, mediates the transport of ceramide from the ER to the *trans*-Golgi regions, where ceramide is converted to sphingomyelin (SM) [5,6]. CERT consists of several functional domains and motifs: an *N*-terminal pleckstrin homology (PH) domain, which preferentially binds phosphatidylinositol 4-monophosphate [PtdIns(4)P]; a serine-repeat motif (SRM); an FFAT (two phenylalanines in an acidic tract) motif, which binds vesicle-associated membrane protein-associated protein (VAP); and a *C*-terminal steroidogenic acute regulatory-related lipid-transfer (START) domain which effectively encloses and transfers ceramide (Figure 1A) [5,7,8].

Previous studies have shown that multiple phosphorylation at the SRM down-regulates CERT activity and SM synthesis, while phosphorylation of serine 132 by protein kinase D (PKD) and the following sequential phosphorylation of the downstream serine/threonine residues by casein kinase 1 γ (CK1G) inhibit both PtdIns(4)P-binding and ceramide-transfer activity [8,9,10,11]. Notably, the phosphoregulatory mechanisms of CERT have recently garnered great attention, because specific missense mutations in the CERT-encoding gene, *CERT1*, were found to be associated with a group of autosomal dominant hereditary intellectual disabilities in humans [12,13,14], and some of these mutations were recently shown to compromise CERT SRM hyperphosphorylation [12,15].

To explain the simultaneous repression of the activities of the PH and START domains, we previously proposed that the hyperphosphorylation of the SRM triggers a conformational change in CERT, which induces mutual masking between the PH domain and the START domain [8]. This proposal was supported by co-crystallography of the purified PH and START domains, which suggests that specific surfaces of the two domains of CERT have significant affinity [16]. In addition, a solution nuclear magnetic resonance (NMR) study suggested that the hyperphosphorylated SRM electrostatically interacts at least partially with the PtdIns(4)P-binding pocket of the CERT PH domain to dissociate the PH domain from the PtdIns(4)P-embedded membrane, thereby possibly facilitating the conformational change in CERT for the PH–START interaction [17]. However, it is not yet clear how such a dramatic structural alteration to attain the PH/START mutual masking can occur in CERT.

Saus and colleagues have suggested that the predominant form of wild-type CERT [also known as Goodpasture antigen-binding protein (GPBP) Δ26] in cells is a homo-trimer [18] and recently identified that the five-residue motif SHCIE in CERT/GPBPΔ26 is involved in the self-interaction of the protein [19] (Figure 1A). Nonetheless, it remains unclear whether the oligomerization of CERT is relevant to its function and its regulation of CERT.

In this study, we hypothesized that a positively charged region serves as a counterpart of the phosphorylated SRM causing a structural change that suppresses the function of CERT and identified a previously uncharacterized cluster of basic amino acids in CERT as a candidate for this hypothetical counterpart. Then, we found that substitution of the basic amino acids in the cluster with alanine reversed the SRM-dependent repression of CERT activities, i.e., SM synthesis, PtdIns(4)P-binding, VAP-binding, ceramide-transfer activity, and localization to the Golgi, while there was partial recovery of SM synthesis activity. Furthermore, we found that alanine substitution moderately destabilized the trimeric status of CERT. These results suggest that the basic amino acid cluster in the coiled-coil region is involved in the regulation of the CERT function.
Figure 1Identification of a putative coiled-coil region in CERT. (**A**) Domains and motifs of CERT. Amino acid sequences are specified for the SRM and CCR in wild-type (WT) and mutants that were used in this study. (**B**) Prediction of coiled-coil regions (CCRs) in CERT. The amino acid residues 268–303 of human CERT (highlighted with a magenta line) were predicted to be in a coiled-coil structure. The Paircoil2 program [20] was used to make the predictions. The decision threshold was set to 0.025 (dashed line), and a sliding window of 21 was used. (**C**) Three-dimensional protein modeling of the CCR in human CERT. CCBuilder 2.0 [21] was used for protein modeling, and PyMol (https://pymol.org/2/ accessed on 14 July 2018) was used to visualize charged surfaces. Basic and acidic amino acid residues are colored in blue and red, respectively. Other residues are colored in white. (**D**) Wheel model of the CCR from wild-type CERT. Lysine/arginine and glutamic acid/aspartic acid are colored in blue and red, respectively. ORIGAMI program [22] was used for wheel model projection.
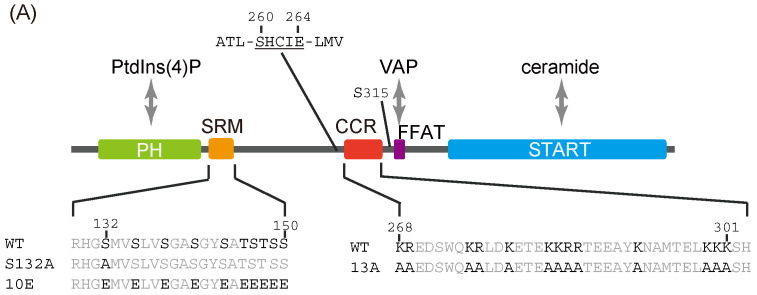

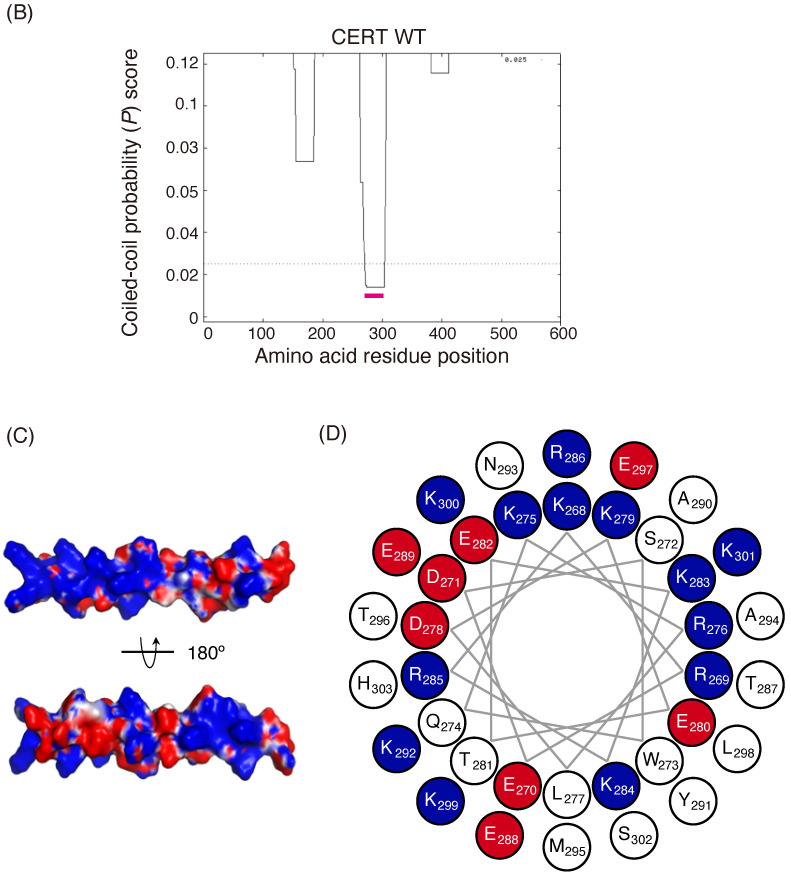


## 2. Results

### 2.1. Identification of a Cluster of Basic Amino Acids in a Putative Coiled-Coil Region in CERT

An α-helical coiled-coil is one of the primary structures that mediates subunit oligomerization in various proteins [23]. We analyzed the entire 598 amino acid sequence of human CERT using the Paircoil2 program and identified a probable coiled-coil region that resides between amino acid residues 266 and 304 (Figure 1B). The predicted coiled-coil region of CERT has a sequence rich in charged amino acids spaced with hydrophobic residues (Figure 1A). We hereafter refer to the putative coiled-coil region as the CCR. A three-dimensional (3D) model of the CCR in CERT was constructed using the CCBuilder 2.0 application, and its surface charges were visualized using the PyMol 2.0 APBS tools (Figure 1C). In this model, the basic residues of the CCR in CERT were predicted to be located on the outer surface of the structure. We hypothesized that the basic amino acid residues exposed on the surface interact electrostatically with hyperphosphorylated SRM and induce conformational changes leading to the inactivation of CERT.

### 2.2. Alanine Substitutions for the Basic Amino Acid Cluster in the CCR Abrogate the Down-Regulatory Effect of SRM Phosphorylation on CERT in Cells

We have previously reported CERT mutants that mimic the hyperphosphorylated (inactive) and dephosphorylated (active) states of SRM, namely CERT 10E and S132A (Figure 1A) [8]. As PKD-dependent phosphorylation of the serine 132 in the SRM induces CK1G-dependent sequential multi-phosphorylation of the downstream serine and threonine residues [10,11], a single base substitution mutation from serine 132-to-alanine can block the inactivation of CERT. CERT 10E has reduced PtdIns(4)P-binding and ceramide-transfer activities compared with wild-type CERT (CERT WT) and CERT S132A. To verify whether the cluster of basic amino acid residues in the CCR is involved in the regulation of CERT activity that depends on SRM phosphorylation/dephosphorylation, we substituted alanine for the 13 basic amino acid residues in the CCR (the substitutions are hereinafter referred to as 13A) (Figure 1A and Appendix A) and generated a 13A single mutant (CERT 13A) and SRM/13A double mutants (CERT 10E/13 A and CERT S132A/13A).

The primary physiological function of CERT is to transport ceramide synthesized in the ER to the *trans*-Golgi regions, where ceramide is converted to SM [1,5]; therefore, we investigated the impact of the 13A mutation on the synthesis of SM in cells. To achieve this aim, we constructed an appropriate cell panel, in which levels of ectopically expressed CERT were adjusted to resemble the endogenous level of wild-type (WT) HeLa cells; this was because the physiological importance of the PH domain and FFAT motif of CERT for de novo synthesis of SM is barely detectable under overexpressing conditions [7]. Even when one copy of CERT cDNA was stably integrated into the host genome using a retroviral vector, the expression levels of the ectopic constructs were found to be considerably beyond the endogenous level. Thus, we employed CERT KO/shCERT cells in which short hairpin RNA for silencing of the CERT mRNA level was stably transformed in CERT KO HeLa cells being the host cells. We obtained multiple transformant clones of CERT KO/shCERT cells stably expressing the CERT WT and various mutants (CERT 10E, S132A, 13A, 10E/13A, and S132A/13A) and constructed a cell panel suitable for our aim (Figure 2B and Appendix A). In the stably transfected cells, about 70% of the CERT WT and about 50% of the CERT 13A were estimated to be hyperphosphorylated in the SRM (Figure 2C and Appendix A), although we could not precisely control the SRM phosphorylation states of CERTs with the wild-type SRM in living cells.

Within the cell panel, de novo SM synthesis was determined by metabolic labeling with [^14^C] serine. The de novo synthesis of SM in the 10E mutant was lower than that in WT cells (Figure 2A), in line with the findings of our previous study [8]. Although the expression level of CERT 10E/13A was lower than that of 10E, the level of SM synthesis tended to be higher in CERT 10E/13A cells than in CERT 10E cells. However, SM synthesis in 10E/13A cells did not reach the levels seen in WT or S132A cells (Figure 2A). Notably, the SM synthesis level in S132A/13A cells was considerably lower than the level in S132A cells, while it was almost equal to the level in 10E/13A cells. These results suggested that the 13A mutation had a negative impact on the highly activated form, CERT S132A, as discussed below. Therefore, the restoration of SM synthesis in 10E/13A cells to the level in S132A/13A cells suggested that the repressive effect of multiple phosphorylation of the SRM (mimicked by the 10E mutation) on the CERT function was abrogated by the 13A mutation.

Using the cell panel, we also examined the subcellular localization of ectopically expressed CERT constructs by indirect immunofluorescence microscopy and determined the Golgi-localization degree of CERT using the degree of colocalization of each CERT mutant with GM130, a Golgi marker. The Golgi-localization degree was determined as the ratio of the fluorescence signal of CERT between Golgi-related regions based on the GM130 signals and in whole cells. Figure 3A shows representative cell images of each CERT mutant and the Golgi-localization degree of CERT. The Golgi-localization degree of each CERT mutant was quantified as the mean ratio of the fluorescent intensity of the CERT signal between the Golgi-related region, which overlaps with the GM130 signal, and the whole cell region (Figure 3B). Compared with CERT S132A, CERT 10E tended to be more abundant in the cytosol, and traces were localized in the Golgi apparatus (Figure 3). Likewise, CERT WT tended to be more abundant in the cytosol, compared with S132A, in line with the finding that approximately 70% of CERT WT in cells exists as its SRM hyperphosphorylated form under our experimental conditions (Figure 2C and Appendix A). The Golgi-localizing degree of CERT 10E/13A was significantly higher than that of CERT 10E and reached the level of CERT S132A (Figure 3). Moreover, the Golgi-localizing degree of CERT 13A was comparable to the level of CERT S132A (Figure 3), although 50% to 60% of CERT 13A was estimated to be hyperphosphorylated in its SRM (Figure 2C and Appendix A), suggesting that not only the dephosphorylated form but also the hyperphosphorylated form of CERT 13A were redistributed to the Golgi apparatus. These results provided additional in vivo evidence that the down-regulating effect of SRM phosphorylation was abrogated by the 13A mutation.

### 2.3. The 13A Mutation Reverses Phosphorylation at Serine 315 and Affinity for VAP in CERT 10E Mutants

The interaction with VAP, an ER-resident transmembrane protein, is required for the ER-to-Golgi ceramide-transfer activity of CERT. A CERT mutant lacking the FFAT motif retains ceramide transport activity and PtdIns(4)P-binding activity, but its de novo SM synthesis is significantly impaired [7]. Phosphorylation of the serine residue at the 315 position (S315) enhances the affinity of the FFAT motif for VAP (Figure 1A) [27]. The phosphorylated form of S315 is detected at more abundant levels in CERT S132A than in CERT 10E, which suggests either activation of phosphorylation or attenuation of dephosphorylation of S315 occurs preferentially when the SRM is de-phosphorylated [27]. We tested whether the phosphorylation status of the S315 and binding affinity for VAP was altered by the 13A mutation. To detect relatively weak interactions between CERT and VAP, we treated cells with a chemical cross-linker prior to cell lysis: *N*-terminal HA-tagged CERT and *N*-terminal FLAG-tagged VAP-A were transiently overexpressed in CERT KO HeLa cells, and the cells were treated with dithiobis (succinimidyl propionate), a chemical cross-linker that is cleavable by reducing agents. Cells were then disrupted with a non-ionic detergent, and the cell lysate was subjected to co-immunoprecipitation with anti-HA or anti-FLAG antibodies. The co-precipitated fraction was dissolved in an SDS loading buffer containing dithiothreitol (DTT) to cleave the bond between CERT and VAP and analyzed by western blotting with antibodies (Figure 4). Compared with CERT WT and 10E, where the phosphorylated form of S315 was not detected, the phosphorylated form of S315 was detected in 10E/13A and S132 A/13A, as well as in S132A (Figure 4). In addition, although only a small quantity of CERT 10E co-precipitated with VAP, the quantity of CERT 10E/13A that co-precipitated was comparable with the levels of the CERT S132A and S132A/13A mutants (Figure 4). These results indicated that even when the SRM is hyperphosphorylated, an additional 13A mutation reverses the status of the S315 phosphorylation and the affinity of CERT for VAP to the levels of the SRM-dephosphorylated CERT. Phosphorylation at the S315 and affinity for VAP was also enhanced in CERT 13A (Figure 4), which suggested that the 13A mutation abrogates various aspects of the SRM hyperphosphorylation (or 10E mutation)-dependent down-regulation in CERT.

In the process of the construction of the 13A mutant, we made two other CCR mutants: All nine lysine/arginine residues between positions 268 and 286 were substituted with alanine in a mutant named 9A, while all four basic residues between positions 289 and 303 were substituted with alanine in another mutant named 4A (Appendix A). Using these new constructs, we examined whether the functional repression of CERT by the 10E mutation in the SRM could be rescued by the 9A and/or 4A mutations. We found that the repression of serine 315 phosphorylation and the VAP-binding of CERT were less reversed by the 9A and the 4A mutation compared with the reversal caused by the 13A mutation (Appendix A). These results suggest that many (even if not all) basic amino acid residues rather than a specific individual residue in the CCR form a counterpart to the multiply phosphorylated SRM in the functional repression of CERT.

### 2.4. The 13A Mutation Restores PtdIns(4)P-Binding and Ceramide-Transfer Activities in CERT 10E

The PtdIns(4)P-binding activity and intermembrane ceramide-transfer activity of CERT are ascribed to the PH domain and START domain, respectively [5]. We examined the PtdIns(4)P-binding and ceramide-transfer activities of various CERT mutants using recombinant proteins expressed in Sf21 insect cells. The SRM of CERT WT may be phosphorylated even in insect cells, which is an uncontrollable event. This means that purified CERT WT does not serve as a fully phosphorylated form nor a non-phosphorylated form in the SRM. Therefore, we did not employ the CERT WT preparation and, instead, we analyzed the effect of 13A mutation in comparison with S132A (which mimics the dephosphorylated form) and 10E (which mimics the fully phosphorylated form). Recombinant human CERT S132A, 10E, S132A/13A, and 10E/13A expressed in Sf21 cells were purified to homogeneity using tandem affinity chromatography (Figure 5A), and the purified proteins were subjected to in vitro assays for PtdIns(4)P-binding and ceramide-transfer activities. CERT 10E/13A preferentially bound to liposomes, including PtdIns(4)P, which was comparable with CERT S132A and S132A/13A (Figure 5B). In the ceramide-transfer assay, the transfer of radioactive long-chain C_16_-ceramide from donor to acceptor liposomes was quantitated. The ceramide-transfer activity of CERT 10E/13A tended to be higher than that of CERT 10E (although this difference was not statistically significant, *p* = 0.072) (Figure 5C). As discussed below, there is the possibility that the 13A mutation itself has a marginally negative impact on the inter-membrane ceramide-transfer activity of CERT. Overall, these results suggest that the CCR mutation reverses the functions of both the PH and START domains of CERT with the 10E mutation, which mimics hyperphosphorylated SRM, to the level of the SRM-dephosphorylated form.

### 2.5. The 13A Mutation Partially Destabilizes the Trimeric State of CERT

As the CCR is located in the vicinity of a region between amino acids 260 and 264, which was previously reported to be required for CERT trimerization (Figure 1A) [19], we assessed the oligomerization states of the four CERT constructs. To this end, the purified proteins were treated with a non-cleavable chemical cross-linker [ethylene glycol bis (succinimidyl succinate); EGS] at various concentrations, separated using SDS-PAGE, and subjected to western blotting analysis. When treated with 0.1–0.8 mM EGS, trimer forms were predominant among cross-linked oligomers for both CERT S132A and CERT 10E (Figure 6A,B and Appendix A), in line with a previous study [19]. Compared with CERT S132A and CERT 10E, their 13A variants (CERT 10E/13A and S132/13A) produced more dimer forms in addition to the non-linked monomer and cross-linked trimer forms (Figure 6B and Appendix A), suggesting that the 13A mutation partially destabilized the CERT trimeric forms.

We also employed blue native PAGE to examine the oligomerization states of the mutants and confirmed that CERT S132A and 10E showed mobilities corresponding mainly to the molecular weights of the homo-trimers, while CERT S132A/13A and 10E/13A migrated like a nearly equivalent mixture of trimers and dimers (Figure 6C). This is consistent with the above results of the cross-linking experiments (Figure 6A,B). Thus, although CERT S132A/13A and 10E/13A retain the ability to homo-trimerize, the 13A mutation might partially destabilize the trimeric state of CERT.

Mysteriously, the mobilities of the monomers of these CERT variants in blue native PAGE were more consistent with the expected mobilities of dimers, rather than those of monomers, when compared with the molecular standard proteins used (Figure 6C). To verify that the lower bands of purified CERT S132A in the blue native PAGE patterns represented monomers, we pre-treated the purified CERT S132A under several different conditions and then analyzed the pre-treated CERT S132A using blue native PAGE (Figure 6D). The pre-exposure to 2% SDS and 100 mM of the reducing reagent DTT almost completely diminished the trimer form and the trimer form converged to the lower band (Figure 6D), indicating that the lower band is the monomer of CERT in blue native PAGE, although its apparent mobility was slower than we expected. The precise reason for this discrepancy is unclear, but a shape-dependent effect might be one cause: CERT is predicted to contain long, unstructured regions (also known as intrinsically disordered regions) [6]. Thus, while the trimers of CERT variants may form compact conformations via inter-molecular interactions, whereas the monomers of CERTs might form relatively extended shapes, thereby moving more slowly than their expected mobilities in size-exclusion chromatography. Pre-exposure of CERT S132A to either 2% SDS or 100 mM DTT resulted in only a partial trimer-to-monomer shift (Figure 6D), suggesting that intermolecular S–S bonding may occur in the CERT trimeric form. It was also noted that when CERT S132A was treated with 100 mM DTT or 4 M urea, an additional band appeared on the blue native PAGE that migrated slightly more slowly than the typical monomer band (Figure 6D). An additional band also appeared for CERT S132A/13A even without the denaturing treatments (Figure 6C). We tentatively assigned these additional bands to the dimeric form of CERTs, but the possibility of being conformational isomers in the monomeric state of CERTs has not been ruled out at present.

## 3. Discussion

Multiple phosphorylation of the CERT SRM results in simultaneous repression of the activities of the *N*-terminal PH domain and *C*-terminal START domain of CERT [8,9]. Interestingly, the repression of the PH domain activity requires the START domain in CERT, while the repression of the START domain activity requires the PH domain, providing a model of SRM phosphorylation-dependent conformational change in CERT [8]. The model is supported by co-crystallography evidence of the purified PH and START domains, which suggests that specific surfaces of the two domains of CERT have significant affinity [16]. However, it is poorly understood how the SRM phosphorylation-dependent conformational change occurs.

In the current study, we identified a previously uncharacterized cluster of lysine/arginine residues, which were predicted to be located on the outer surface of a probable coiled-coil fold (Figure 1). Analysis of PtdIns(4)P-binding activity, VAP-binding activity, S315 phosphorylation, and Golgi localization (Figure 3, Figure 4 and Figure 5B) demonstrated that the negative effects of the 10E mutation on these activities were reversed near to the positive control level (the level of CERT S132A) by alanine substitutions (i.e., the 13A mutation) of the basic amino acids in the cluster. On the other hand, the effects of the 13A mutation on the de novo synthesis of SM in intact cells and the ceramide-transfer activity in a cell-free assay system were complicated: The de novo synthesis of SM activity in CERT 10E/13A cells was similar to (but tended to be higher than) that in CERT 10E cells but never reached the level seen in CERT S132A cells (Figure 2A), and the level of the ceramide-transfer activity of CERT 10E/13A did not reach the level of CERT S132A (Figure 5C). Saus and colleagues showed that the predominant form of wild-type CERT (also known as GPBPΔ26) in cells is a homo-trimer [18,19], which is supported by our present study (Figure 6). The five-residue motif SHCIE, which is located in the vicinity of the CCR (Figure 1A), was shown to be crucial for the trimerization of CERT [19], and our present study suggested that the 13A mutation partially destabilized the trimer organization of CERT (Figure 6). The 13A mutation itself may have a marginally negative impact on intermembrane ceramide-transfer activity via partially destabilizing the trimeric state of CERT, and, therefore, the CERT 10E/13A construct could not fully reverse its SM synthesis and ceramide-transfer activities to the CERT S132A level. Taking these findings together, we conclude that the negative impact of SRM hyperphosphorylation on the function of CERT is reversed by the 13A mutation, albeit not perfectly, and that the basic amino acid cluster of the CCR is involved in the SRM phosphorylation-dependent repression of CERT.

Based on these results in the present and previous studies, we propose an updated model to explain how multiple phosphorylation of the SRM represses the function of CERT (Figure 7). This model suggests that the positively charged surface of the CCR interacts with the highly phosphorylated SRM and that the 13A mutation loses this electrostatic interaction. Consequently, CERT 10E/13A, similar to the constitutively activated CERT S132A, may form an “open structure” in which the PH and START domains are more accessible to their lipid ligands, i.e., PtdIns(4)P and ceramide, respectively (Figure 7). In addition, the open structure facilitates the FFAT motif to interact with VAP, which is an important step to exert CERT-mediated transport of ceramide at the ER–Golgi contact sites [7,27]. However, it should be pointed out that our model does not determine whether the basic amino acid cluster in the CCR and the phosphorylated SRM interact intramolecularly (i.e., in the same CERT molecule) or in an intermolecular manner (between different CERT molecules in one trimer, analogous to “braided hair”). Structural biological studies of the native complex of the full-size CERT will be needed to further elucidate how mutual masking between the PH and START domains in CERT occurs. It should also be emphasized that we do not deny the possibility that the oligomerizing nature of CERT in intact cells may be more dynamic than the trimeric model, which is derived mainly from static “snapshot” data. Although we putatively depicted that all FFAT motifs and PH domains of one CERT trimer are simultaneously interacting with VAP and PtdIns(4)P, respectively, in the model (Figure 7), the binding of one or two FFAT motifs and PH domains in the CERT trimer to their partners may be enough for CERT to act at the ER–Golgi contact site.

Several missense mutations in human *CERT1* have been reported to be associated with inherited intellectual disorders (IDs). ID-associated amino acid substitutions are enriched in the SRM region [12,13,14,28,29], and these substitutions likely abrogate the SRM phosphorylation-dependent inactivation of CERT [12]. In addition, some ID-associated substitutions occur outside the SRM, i.e., T166A, F182L, and G243R substitutions in CERT [30,31]. A recent study demonstrated that CERT G243R exhibits a constitutively activated phenotype [15]. Although none of T166A, F182L, or G243R are constituents of the CCR (which is defined from K268 to K301), they may also abrogate the SRM-dependent regulatory system in CERT. If ID-associated mutations in *CERT1* are mapped in the CCR in the future, it should provide strong evidence for the involvement of the CCR in human health-relevant functional regulation of CERT.

## 4. Materials and Methods

### 4.1. Materials

Phosphatidylcholine (more specifically, 1-palmitoyl-2-oleoyl-*sn*-glycero-3-phosphocholine), phosphatidylethanolamine (more specifically, 1-palmitoyl-2-oleoyl-*sn*-glycero-3-phosphoethanolamine), and lactosylceramide were obtained from Avanti Polar Lipids. Phosphatidylinositol 3-monophosphate [PtdIns(3)P] and PtdIns(4)P were obtained from Cayman Chemicals. *Ricinus communis* agglutinin was obtained from Vector Laboratories, Inc. Phosphate-buffered saline (PBS) was obtained from Fujifilm Wako Pure Chemical Co.

### 4.2. Antibodies and Plasmids

Anti-HA antibody (#3F10) was obtained from Roche Diagnostics; anti-His (#D291-3S) from Medical & Biological Laboratories Co., Ltd.; anti-CERT from Abcam (#ab72536); anti-GM130 (#610823) from BD Biosciences; and anti-GAPDH (#016-25523) from Fujifilm Wako Pure Chemical Co. The production of anti-VAP-A chicken polyclonal antibody and anti-phospho-serine 315 antiserum was described previously [27].

The construction of the CERT S132A and 10E mutants was described previously [8]. The *N*-terminal HA-tagged WT CERT cDNA in pBlueScript II SK (+) (pBS/nHAcFL-CERT WT) was used as a template to mutagenize the thirteen lysine/arginine residues in the CCR (amino acids 268–303) to alanine (CERT 13A). The CCR was divided into two overlapping sub-regions, CC1 (amino acids 268–291, CERT 9A) and CC2 (287–303, CERT 4A), and the following two sets of oligonucleotides were used for mutagenesis, respectively. CC1: 5′-GAACTAATGGTTGCAGCTGAGGACAGCTGGCAGGCGGCCCTGGATGCGGAAACTGAGGCGGCAGCAGCAACAGAGGAAGCATAT-3′ and 5′-TGCTTCCTCTGTTGCTGCTGCCGCCTCAGTTTCCGCATCCAGGGCCGCCTGCCAGCTGTCCTCAGCTGCAACCATTAGTTCAAT-3′, CC2: 5′-ACAGAGGAAGCATATGCAAATGCAATGACAGAACTTGCGGCAGCATCCCACTTTGGAGGACCAGAT-3′ and 5′-TGGTCCTCCAAAGTGGGATGCTGCCGCAAGTTCTGTCATTGCATTTGCATATGCTTCCTCTG-3′. WT and mutant CERT cDNAs were transferred to pMXs-IRES-Bsd vector [32] or pENTR/D-TOPO vector (Thermo Fisher Scientific) for expression in mammal cells or insect cells, respectively.

A retrovirus-based CERT shRNA expression vector (shCERTv1) was constructed by annealing the following two oligonucleotides and cloning the annealed nucleotides into pSilencer v 5.1 vector (Thermofisher). shCERT-pSilencer v1 sense: 5′-GATCCGCGAGAGTATCCTAAATTTAAGTTCTCTAAATTTAGGATACTCTCGCTTTTTTGGAAA-3′ and shCERT-pSilencer v1 antisense: 5′-AGCTTTTCCAAAAAAGCGAGAGTATCCTAAATTTAGAGAACTTAAATTTAGGATACTCTCGCG-3′.

### 4.3. Cell Culture and Transfections

HeLa cells expressing a mouse ecotropic retroviral receptor, HeLa-mCAT#8 cells [33], were cultured in DMEM with 10% (*v/v*) FBS at 37 °C with 5% CO_2_. A *CERT*-disrupted mutant HeLa cell line (clone name, TAL-CE#14 clone) that was previously established in our laboratory [34] was used as CERT KO HeLa cells. For the establishment of CERT KO/shCERT, the CERT KO cells were transduced with virus particles produced using the shCERTv1 plasmid. An isolated clone (TAL-CE#14-shCE#1 clone) was transduced with virus particles produced using pMXs-IRES-Bsd expression vectors [32] encoding *N*-terminal HA-tagged human CERT S132A, CERT 10E, CERT 13A, CERT S132A/13A, or CERT 10E/13A. Sf21 cells (Thermo Fisher Scientific, Waltham, MA, USA) were cultured in SF900-II medium in monolayer or suspension at 27 °C. Cell transfection was performed as described below.

### 4.4. Baculovirus Expression and Purification

WT and mutant CERT cDNAs cloned in pENTR/D-TOPO vector (Thermo Fisher Scientific) were transferred into BaculoDirect linear DNA (Thermo Fisher Scientific, Waltham, MA, USA) by recombination, and the resulting ligation mixtures were transfected to Sf21 cells in monolayer using Cellfectin II (Thermo Fisher Scientific, Waltham, MA, USA). High-titer viral stocks were generated under selection with 100 μM ganciclovir (Tokyo Chemical Industry, Tokyo, Japan), transduced into Sf21 cells in suspension, and cultured for 72 h at 27 °C. Cells were harvested, lysed, and purified using an anti-His-tag and an anti-HA-tag affinity resin. Briefly, cells were resuspended in TALON buffer [50 mM HEPES-NaOH (pH 7.5), 300 mM NaCl] containing 30 mM imidazole and protease inhibitor cocktail (complete EDTA-free, Roche, Basel, Switzerland) and disrupted by passing through 18-gauge and 25-gauge needles followed by sonication with a probe-type sonicator (UP50H, Dr. Hielscher GmbH, Chamerau, Germany). Cell lysates were collected by centrifugation at 100,000× *g* for 1 h at 4 °C, then loaded on a TALON metal affinity column (Clontech Laboratories, Inc., Mountain View, CA, USA). The *N*-terminal His-tagged CERT proteins were on-column treated with λ protein phosphatase (NEB) for 1 h at 4 °C, then eluted with TALON buffer containing 150 mM imidazole. The eluent was subjected to concentration and buffer exchange [50 mM HEPES-NaOH (pH 7.5), 500 mM NaCl] using a 30-kDa cutoff Amicon Ultra column (Merck Millipore, Burlington, MA, USA), bound to anti-HA affinity resin (Merck Millipore, Burlington, MA, USA) o/n at 4 °C, and eluted with a buffer [50 mM HEPES-NaOH (pH 7.5), 500 mM NaCl] containing 1 μM HA peptide. Aliquots of the eluent were stored at −80 °C.

### 4.5. In Vitro PtdIns(4)-Binding and Ceramide-Transfer Assays

PtdIns(4)P-binding by purified recombinant CERT proteins was measured using liposomes without or with 0.5 mol% of PtdIns(3)P or PtdIns(4)P, as previously described but with some modifications [8]. Liposomes composed of phosphatidylcholine:phosphatidylethanolamine:lactosylceramide:phosphoinositide (15.87:4:6:0.13, mol/mol) were prepared by sonication using a probe-type sonicator (UP50H, Dr. Hielscher GmbH, Chamerau, Germany) and pre-centrifuged at 20,400× *g* for 5 min at 4 °C. Purified CERT was pre-centrifuged at 100,000× *g* for 30 min at 4 °C. Liposomes containing in total 26 nmol of lipids were incubated with 200 fmol of recombinant CERT in 50 μL of buffer C containing 0.3 mg/mL BSA and 0.1 (*v/v*) CHAPS for 30 min at 4 °C. *Ricinus communis* agglutinin (18.75 μg) was added to terminate the reaction, incubated on ice for 10 min to aggregate the liposomes, and centrifuged at 20,400× *g* for 1 min at 4 °C. PtdIns(4)P-bound (pellet fraction) or non-bound (supernatant fraction) CERT was quantified by immunoblotting analysis using an anti-HA antibody.

Ceramide-transfer between artificial donor and acceptor liposomes was measured as described elsewhere, with some modifications [5,35]. Briefly, donor and acceptor liposomes composed of phosphatidylcholine:phosphatidylethanolamine:lactosylceramide:[^14^C]ceramide (64:16:40:1, mol/mol), and phosphatidylcholine: phosphatidylethanolamine (320:80, mol/mol), respectively, were prepared by sonication using a probe-type sonicator (UP50H, Dr. Hielscher Gmb) and pre-centrifuged at 20,400× *g* for 1 min at 4 °C to remove lipid aggregates. Purified CERT was pre-centrifuged at 100,000× *g* for 30 min at 4 °C to exclude aggregated protein. Acceptor liposomes containing a total 400 nmol of lipids were incubated with 2 pmol of recombinant CERT in 80 μL of buffer C [20 mM HEPES-NaOH (pH 7.5), 50 mM NaCl, 1 mM EDTA] on ice. Donor liposomes (20 μL) were added to the assay mixtures to start the reactions and incubated for 10 min at 37 °C. The reactions were terminated by adding 37.5 μg of *Ricinus communis* agglutinin, incubated for 15 min on ice, and the donor liposomes were precipitated by centrifugation at 20,400× *g* for 3 min at 4 °C. The radioactivity of the supernatant including the acceptor liposomes was measured using liquid scintillation counting.

### 4.6. Chemical Cross-Linking

Recombinant CERT proteins [10.8 ng of protein in 30 μL of 50 mM HEPES-NaOH (pH 7.4)] were incubated with or without various concentrations of ethylene glycol bis (succinimidyl succinate) (EGS, Thermo Fisher Scientific, Waltham, MA, USA) on ice for 2 h. After addition of Tris-HCl (pH 7.4) to the mixture at a final concentration of 47.6 mM, the resultant mixture was incubated for 15 min on ice to terminate the cross-linking reaction. The terminated mixture was diluted in SDS-PAGE loading buffer and subjected to western blotting analysis.

### 4.7. Blue Native PAGE

Samples were prepared using the NativePAGE^TM^ sample preparation kit (Thermo Fisher Scientific, Waltham, MA, USA), according to the manufacturer’s instructions. Briefly, purified recombinant CERT samples and HMW Native Marker Kit (GE Healthcare, Chicago, IL, USA) were diluted in native PAGE buffer supplemented with 1% digitonin, and CBB G-250 was added to a final concentration of 0.25%. Samples were separated on a NativePAGE™ Novex^®^ 4–16% Bis-Tris gel (Thermo Fisher Scientific, Waltham, MA, USA), and the gel was incubated in 25 mM Tris-glycine buffer (pH 8.3) with 0.1% SDS for 15 min at room temperature with shaking. Proteins were transferred to a PVDF membrane (Bio-Rad, Hercules, CA, USA), and the membrane was fixed in 10% acetic acid for 15 min, rinsed with distilled water, and air-dried. The membrane was de-stained in methanol, rinsed with distilled water, and subjected to western blotting analysis.

### 4.8. Immunoblotting and Co-Immunoprecipitation

Protein samples were prepared in NuPAGE lithium dodecyl sulfate sample buffer (Thermo Fisher Scientific) with 100 mM DTT and heated for 5 min at 70 °C. Samples were resolved by SDS-PAGE and transferred to PVDF membranes (Bio-Rad). Proteins were visualized by incubation with primary antibodies, which was followed by incubation with secondary HRP-conjugated antibodies for detection using a LuminoGraph II system (ATTO Corporation). Precision Plus Protein All Blue Standards (Bio-Rad) were used as molecular standards for SDS-PAGE.

Co-immunoprecipitation of CERT and VAP-A was carried out as previously described, with some modifications [27]. Briefly, CERT KO cells were transiently co-transfected with pcDNAneo/nHA-hCERT and pcDNAhyg/nFL-hVAP-A in 6-well plates and cultured for 24 h at 37 °C. Cells were washed twice with PBS and incubated on ice in PBS containing a cross-linker [2 mM dithiobis (succinimidyl propionate)] (Thermo Fisher Scientific, Waltham, MA, USA) for 30 min. The cross-linking reaction was terminated by adding 50 mM Tris-HCl (pH 7.4) buffer, followed by 10 min incubation on ice. Cells were washed twice with PBS and lysed on ice in buffer A [50 mM Tris-HCl (pH 7.4), 1 mM EDTA, 1 mM EGTA, 100 mM NaCl, 50 mM NaF, 5 mM Na-pyrophosphate, 10 mM Na-β glycerophosphate] containing 1% (*v/v*) Triton X-100 and supplemented with a protease inhibitor cocktail (cOmplete EDTA-free, Roche) of PPI-2 and PPI-3 phosphatase inhibitors (Merck Millipore). Lysates were precleared by centrifugation at 14,000× *g* for 10 min at 4 °C, and the protein concentrations of the precleared lysate fractions were determined using the BCA protein assay kit (Thermo Fisher Scientific, Waltham, MA, USA). HA-CERT or FLAG-VAP was immunoprecipitated from supernatants containing equivalent amounts of total protein using an anti-HA agarose gel (Merck Millipore, Burlington, MA, USA) or an anti-FLAG M2 agarose gel (Sigma-Aldrich, St. Louis, MO, USA), washed twice with buffer A containing 0.1% (*v/v*) Triton X-100, and resuspended in NuPAGE lithium dodecyl sulfate sample buffer (Thermo Fisher Scientific, Waltham, MA, USA). Immunoprecipitates were subjected to SDS-PAGE and immunoblotting analysis using appropriate antibodies.

### 4.9. Immunofluorescence Microscopy

Cells cultured on glass coverslips (Matsunami, Tokyo, Japan) were fixed in Mildform^®^ 10N (Fujifilm Wako Pure Chemical Co., Osaka, Japan) for 10 min at room temperature, permeabilized in 0.1% (*v/v*) Triton X-100/PBS for 10 min at room temperature, blocked in 3% (*v/v*) BSA/PBS, and incubated with antibodies diluted in 0.1% (*v/v*) BSA/PBS. The cover slips were rinsed with distilled water and mounted with Fluoromount (Diagnostic BioSystems, Pleasanton, CA, USA). Fluorescence images were obtained with a fluorescence microscope (BZ-X710 with a 60× Plan-Apochromat V NA 1.20 objective lens, Keyence).

### 4.10. Quantification of the Golgi-Localization Degree of CERT Fluorescence Signals

To quantify the Golgi-localization degree of each CERT mutant, the mean ratio of the fluorescence intensity of the CERT signal between the Golgi-related region and the whole cell region was determined using Fiji/ImageJ software (NIH) [36]. For segmentation of the Golgi-related regions, raw fluorescence images of GM130 were processed by the automatic thresholding Otsu method, and the binary images obtained were further processed by “Dilate” processing. The whole cell regions were segmented manually. The mean ratio of the fluorescence intensity of CERT in the two segmented regions in each cell was calculated.

## Figures and Tables

**Figure 2 ijms-23-08576-f002:**
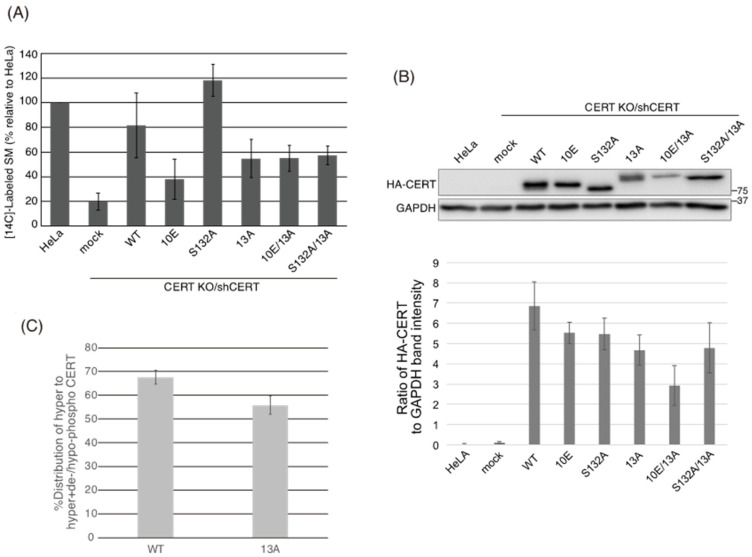
Alanine substitutions for the basic amino acid cluster in the CCR abrogate the down-regulatory effect of the CERT 10E mutation on the de novo synthesis of SM. (**A**) The indicated CERT constructs (please note, an empty vector was used for the mock situation) were stably expressed in HeLa CERT KO/shCERT cells. “HeLa” represents HeLa/mCAT-1 cells that were used as the wild-type control HeLa cells. SM synthesis in these cells was measured by [^14^C]serine incorporation after incubation for 24 h. Results are the mean and SEM of three experiments. (**B**) Total cell lysates from the indicated cells were analyzed by western blotting with anti-HA and anti-GAPDH antibodies. The ratios of HA-CERT band intensity to GAPDH band intensity calculated from three experiments are shown as bar graphs. (**C**) The signal intensity of the bands corresponding to the hyper-phosphorylated and de-/hypo-phosphorylated forms of CERT WT and CERT 13A was quantified using ImageJ software [24], and the percentage distribution values of their hyperphosphorylated forms were estimated from three independent experiments, as described in the legend to Appendix A.

**Figure 3 ijms-23-08576-f003:**
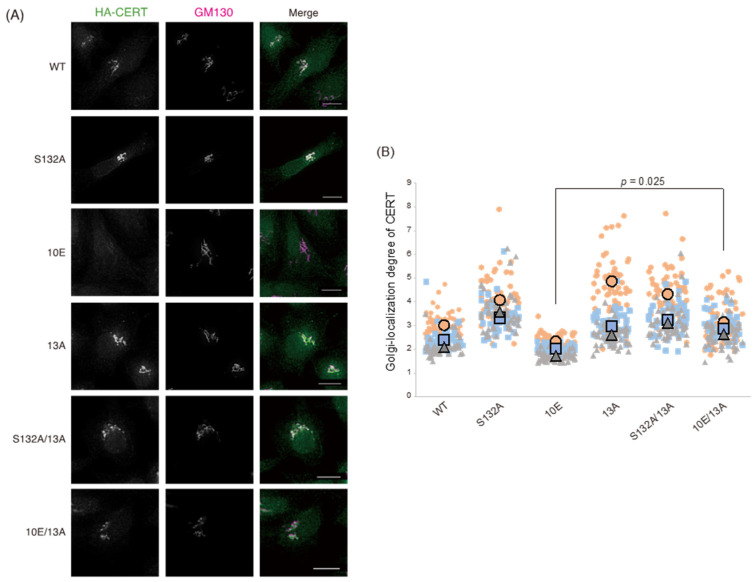
The 13A mutation enhances localization of CERT 10E to the Golgi apparatus. (**A**) HeLa CERT KO/shCERT cells expressing HA-CERT WT, S132A, 10E, 13A, S132A/13A, or 10E/13A were co-immunostained with an anti-HA rat monoclonal antibody and an anti-GM130 mouse monoclonal antibody. All scale bars represent 10 μm. (**B**) The Golgi-localization degree of each CERT mutant was determined as described in the Materials and Methods section. Results are shown in SuperPlots [25]. The small symbols indicate the analyzed values of all cells in each experiment, and the large symbols indicate the means of the values in each experiment. Results of three experiments are shown in orange circles, blue squares, and gray triangles, respectively. Significance was determined using the paired *t*-test adjusted by the Benjamini–Hochberg method [26] for multiple comparisons; *p*-values < 0.05 were considered statistically significant.

**Figure 4 ijms-23-08576-f004:**
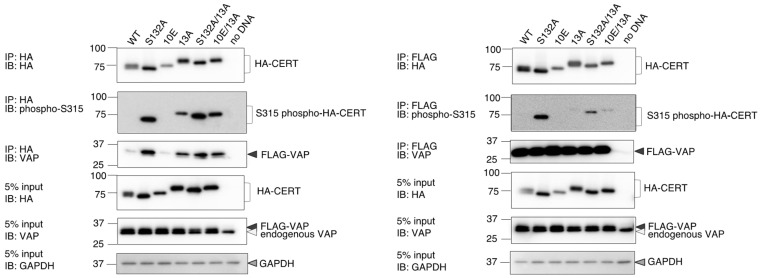
The 13A mutation enhances phosphorylation at the serine 315 and affinity of CERT for VAP. Various constructs of HA-CERT (WT, S132A, 10E, S132A/13A, and 10E/13A) and FLAG-VAP-A were transiently expressed in HeLa CERT KO cells, cross-linked, lysed, and co-immunoprecipitated (IP) using anti-HA antibody- or anti-FLAG M2-conjugated agarose gel. The immunoprecipitate and the 5% input samples were separated using SDS-PAGE and immunoblotted with anti-HA, anti-VAP-A, anti-phospho-serine 315, and anti-GAPDH antibodies.

**Figure 5 ijms-23-08576-f005:**
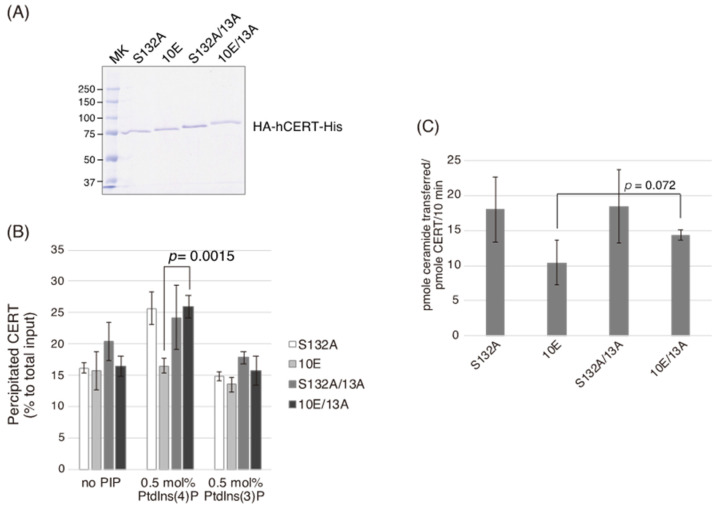
The 13A mutation restores the ceramide-transfer and PtdIns(4)P-binding activities of CERT 10E. (**A**) Quantities (0.5 μg) of recombinant CERT S132A, 10E, S132A/13A, and 10E/13A proteins purified from Sf21 cells were resolved by SDS-PAGE and stained with Coomassie brilliant blue. (**B**) The binding of purified WT and CERT mutants to liposomes without or with 0.5 mol% PtdIns(3)P or PtdIns(4)P was determined by quantification of their distribution in the supernatant and pellet fractions, respectively. Results shown are the means and SEM of three experiments. Significance was determined using a two-tailed Student’s *t*-test; *p*-values < 0.05 were considered statistically significant. (**C**) The transfer of [^14^C]ceramide from donor to acceptor liposomes by CERT was measured. Results shown are the means and SEM of four experiments. Significance was determined using a two-tailed Student’s *t*-test; *p*-values < 0.05 were considered statistically significant.

**Figure 6 ijms-23-08576-f006:**
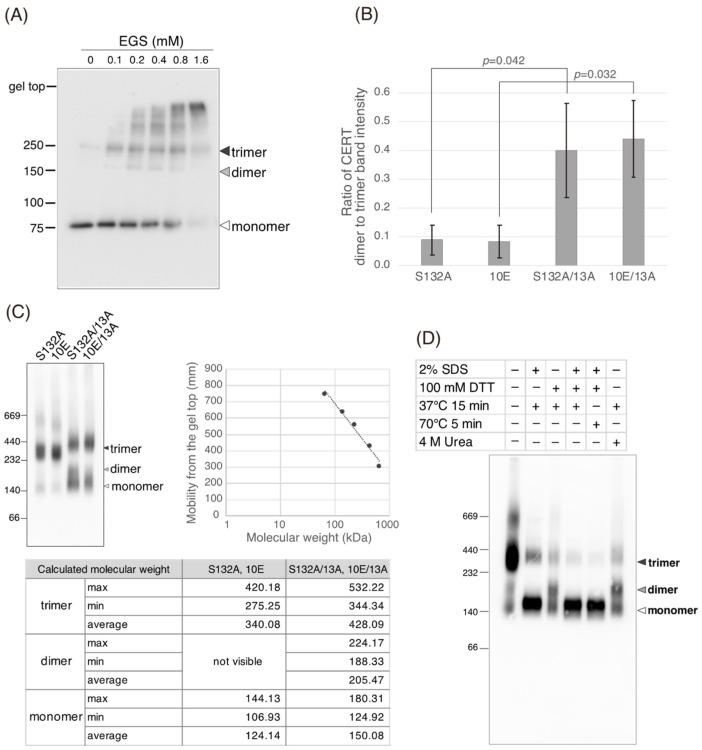
The 13A mutation partially destabilizes the trimeric state of CERT. (**A**) Recombinant CERT S132A, 10E, S132A/13A, and 10E were incubated with the vehicle solvent (DMSO) or EGS (0.1, 0.2, 0.4, 0.8, 1.6 mM) and, after terminating the reaction, aliquots (5 ng of protein) of the samples were subjected to western blotting analysis with an HRP-conjugated anti-HA rat monoclonal antibody. The western blotting image of S132A is shown as a representative example in the figure. See Appendix A for all western blotting images. The acrylamide concentration used for the gel of SDS-PAGE was 6%. (**B**) The ratios of dimer to trimer CERT band intensity calculated from the experiment using 0.8 mM EGS in panel A and Appendix A are shown as bar graphs. Significance was determined using a two-tailed Student’s *t*-test; *p*-values < 0.05 were considered statistically significant. (**C**) Quantities (10 ng) of recombinant CERT S132A, 10E, S132A/13A, and 10E/13A proteins were separated using blue native PAGE and detected with an HRP-conjugated anti-HA antibody. A calibration curve was drawn based on the molecular weight of each band of the marker and its mobility on blue native PAGE to quantify the molecular weight of each band in the recombinant CERT protein. Quantitative values are shown in the table. (**D**) Recombinant CERT S132A protein was incubated under various denaturing conditions, separated by blue native PAGE, and detected by western blotting analysis using an anti-HA antibody.

**Figure 7 ijms-23-08576-f007:**
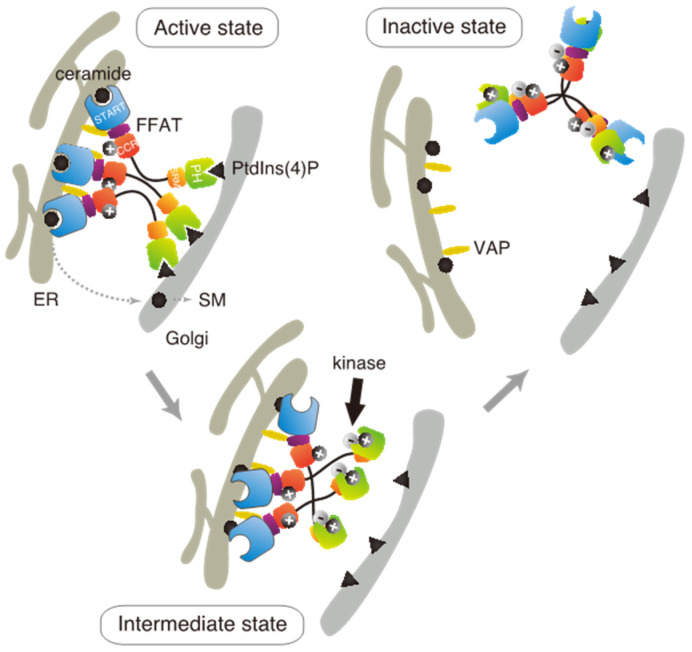
A model for CERT-inactivation that is dependent on SRM phosphorylation. Trimeric CERT is illustrated as a predominant oligomeric form of CERT in cells, although oligomerization states of CERT might change dynamically among monomer, dimer, trimer, and other larger oligomers. CERT transfers ceramide at the ER–Golgi membrane contact site. Ceramide transferred to the Golgi is enzymatically converted to SM. Kinases (PKD and CK1G) phosphorylate CERT at the SRM, resulting in it bearing a negative charge. The negatively charged SRM can initially interact electrostatically with the positively charged region of the PH domain [17] and then with the positively charged surface of the CCR. This interaction drives the inter-masking of the PH and START domains, which results in the inactivation of CERT.

## Data Availability

Raw data obtained in this study are available from the generalist repository, figshare (doi: 10.6084/m9.figshare.14787963), or from the author upon reasonable request.

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
