# Peer review of "Involvement of a Cluster of Basic Amino Acids in Phosphorylation-Dependent Functional Repression of the Ceramide Transport Protein CERT"

_ijms, 2022, doi:10.3390/ijms23158576_

Round 1
Reviewer 1 Report
The manuscript presents high-quality research into the regulation of the function of CERT. However, this reviewer is concerned with the large change in the charge of CERT by the simultaneous replacement of 13 positively-charged residues with small non-polar alanine residues. The large change in the charge of CERT could be responsible for the change in function. For instance, the positive charge that was eliminated could have contributed to the dissociation of the trimer by electrostatic repulsion. Perhaps similar results would have been obtained by generating a similar change in overall charge at some other location in the CERT structure. Thus, it is possible that the region identified by the authors may not be directly involved in the physiological regulation of CERT. Also, one wonders whether a more modest modification of say 6 residues would have resulted in a similar charge in function.
Not wanting to delay the publication of this very nice study, I just ask the authors to consider the issues raised as they continue this line of work.
Author Response
Comments from Reviewer 1:
The manuscript presents high-quality research into the regulation of the function of CERT. However, this reviewer is concerned with the large change in the charge of CERT by the simultaneous replacement of 13 positively-charged residues with small non-polar alanine residues. The large change in the charge of CERT could be responsible for the change in function. For instance, the positive charge that was eliminated could have contributed to the dissociation of the trimer by electrostatic repulsion. Perhaps similar results would have been obtained by generating a similar change in overall charge at some other location in the CERT structure. Thus, it is possible that the region identified by the authors may not be directly involved in the physiological regulation of CERT. Also, one wonders whether a more modest modification of say 6 residues would have resulted in a similar charge in function.
Not wanting to delay the publication of this very nice study, I just ask the authors to consider the issues raised as they continue this line of work.
Response to Reviewer 1:
Thank you very much for your favorable assessment of our manuscript. About the reviewer’s comment “whether a more modest modification of say 6 residues would have resulted in a similar charge in function”, we had a similar question. Therefore, we performed some additional experiments, the results of which are presented in Figure S4 and briefly explained in the revised manuscript as follows (Lines 273-283). “In the process of the construction of the 13A mutant, we made two other CCR mutants: All nine lysine/arginine residues between positions 268 and 286 were substituted with alanine in a mutant named 9A, while all four basic residues between positions 289 and 303 were substituted with alanine in another mutant named 4A (Figure S4). Using these new constructs, we examined whether the functional repression of CERT by the 10E mutation in the SRM could be rescued by the 9A and/or 4A mutations. We found that the repression of serine 315 phosphorylation and the VAP-binding of CERT were less reversed by the 9A and the 4A mutation compared with the reversal caused by the 13A mutation. These results suggest that many (even if not all) basic amino acid residues rather than a specific individual residue in the CCR form a counterpart to the multiply phosphorylated SRM in the functional repression of CERT”
In addition, a wheel model of the wild-type CERT has been added to Figure 1 to visually explain more clearly that many basic amino acid residues are located on the surface of the coiled-coil structure.
Reviewer 2 Report
The manuscript (MS) entitled “Involvement of a cluster of basic amino acids in phosphorylation-dependent functional repression of the ceramide transport protein CERT” by Goto et., all showing some interesting findings about CERT functional mechanism. There are few concerns are needed to address before final publication.
1. English language requires grammar editing in the MS at some places.
2. Introduction is little short that can be improved by highlighting the important findings.
3. Immunofluorescence (ICC) images in figure 3 are not very clear. Authors need to provide high resolution and magnification images.
4. It would be better if authors could provide any in-vivo data to strengthen their obtained results.
5. Discussion part is not easy to follow and it could be improved by discussing the importance of results keeping other as a reference.
Author Response
Comments from Reviewer 2:
Point 1. English language requires grammar editing in the MS at some places.
Point 2. Introduction is little short that can be improved by highlighting the important findings.
Point 3. Immunofluorescence (ICC) images in figure 3 are not very clear. Authors need to provide
high resolution and magnification images.
Point 4. It would be better if authors could provide any in-vivo data to strengthen their obtained
results.
Point 5. Discussion part is not easy to follow and it could be improved by discussing the importance
of results keeping other as a reference.
Response to Reviewer 2:
Thank you very much for your critical and constructive comments. According to your comments, we have made improvements to our manuscript as outlined below.
Point 1. We have reviewed the entire document and made grammatical corrections using track changes.
Point 2. Lines 36-83: To make the Introduction easier to read, we divided it into several paragraphs. Also, more information about the background to this study has been added, as follows: “Saus and colleagues have suggested that the predominant form of wild-type CERT [also known as Goodpasture antigen-binding protein (GPBP) Δ26] in cells is a homo-trimer [18] and recently identified that the five-residue motif SHCIE in CERT/GPBPΔ26 is involved in the self-interaction of the protein [19] (Figure 1A). Nonetheless, it remains unclear whether the oligomerization of CERT is relevant to its function and its regulation of CERT.”
The final paragraph of the Introduction has been rewritten as follows: “In this study, we hypothesized that a positively charged region serves as a counterpart to the phosphorylated SRM, causing a structural change that suppresses the function of CERT, and identified a previously uncharacterized cluster of basic amino acids in CERT as a candidate for this hypothetical counterpart. Then, we found that substitution of the basic amino acids in the cluster with alanine reversed the SRM-dependent repression of CERT activities, i.e., SM synthesis, PtdIns(4)P-binding, VAP-binding, ceramide-transfer activity, and localization to the Golgi, while there was partial recovery of SM synthesis activity. Furthermore, we found that alanine substitution moderately destabilized the trimeric status of CERT. These results suggest that the basic amino acid cluster in the coiled-coil region is involved in the regulation of the CERT function.”
Point 3. We have uploaded a pdf file containing high-resolution and -magnification images for Figure 3A.
Point 4. In Figure 2A, we examined whether the ectopic expression of various CERT mutants in the endogenous CERT-disrupted cells rescues the synthesis of sphingomyelin in living cells. The results are in line with our main conclusion, that the basic amino acid cluster in the CCR is involved in the SRM phosphorylation-dependent repression of CERT, although a negative impact of the 13A mutation on the intermembrane transfer of ceramide may occur, as discussed. We believe that Figure 2A will meet the reviewer’s request because it represents a set of in vivo data.
Point 5. To make the Discussion easier to follow, we have rewritten the main part of this section, as shown below (Lines 421-449).
“In the current study, we identified a previously uncharacterized cluster of lysine/arginine residues, which was predicted to be located on the outer surface of a probable coiled-coil fold (Figure 1). Analysis of PtdIns(4)P-binding activity, VAP-binding activity, S315 phosphorylation, and Golgi localization (Figures 3, 4, 5B) demonstrated that the negative effects of the 10E mutation on these activities were reversed near to the positive control level (the level of CERT S132A) by alanine substitutions (i.e., the 13A mutation) of the basic amino acids in the cluster. On the other hand, the effects of the 13A mutation on the de novo synthesis of SM in intact cells and the ceramide-transfer activity in a cell-free assay system were complicated: The de novo synthesis of SM activity in CERT 10E/13A cells was similar to (but tended to be higher than) that in CERT 10E cells but never reached the level seen in CERT S132A cells (Figure 2A), and the level of the ceramide-transfer activity of CERT 10E/13A did not reach the level of CERT S132A (Figure 5C). Saus and colleagues showed that the predominant form of wild-type CERT (also known as GPBPΔ26) in cells is a homo-trimer [18,19], which is supported by our present study (Figure 6). The five-residue motif SHCIE, which is located in the vicinity of the CCR (Figure 1A), was shown to be crucial for the trimerization of CERT [19], and our present study suggested that the 13A mutation partially destabilized the trimer organization of CERT (Figure 6). The 13A mutation itself may have a marginally negative impact on intermembrane ceramide-transfer activity, via partially destabilizing the trimeric state of CERT, and, therefore, the CERT 10E/13A construct could not fully reverse its SM synthesis and ceramide-transfer activities to the CERT S132A level. Taking these findings together, we conclude that the negative impact of SRM hyperphosphorylation on the function of CERT is reversed by the 13A mutation, albeit not perfectly, and that the basic amino acid cluster of the CCR is involved in the SRM phosphorylation-dependent repression of CERT.”